# Fabrication and Calibration of Pt-Rh10/Pt Thin-Film Thermocouple

**DOI:** 10.3390/mi14010004

**Published:** 2022-12-20

**Authors:** Fengxiang Wang, Zhenyu Lin, Zhijie Zhang, Yanfeng Li, Haoze Chen, Jiaqi Liu, Chao Li

**Affiliations:** 1School of Instrument and Electronics, North University of China, Taiyuan 030051, China; 2Key Laboratory of Instrumentation Science & Dynamic Measurement, North University of China, Taiyuan 030051, China; 3School of Electronic and Control Engineering, North China Institute of Aerospace Engineering, Langfang 065000, China

**Keywords:** Pt-Rh10/Pt, thin-film thermocouple, screen printing process, static calibration, dynamic response

## Abstract

Aiming at the dynamic testing of the ignition temperature of micro-initiating explosives, a novel Pt-Rh10/Pt thin-film thermocouple was designed in this paper. The author carried out the preparation of the thermocouple by using a screen printing process on an Al_2_O_3_ ceramic substrate. The formed thermocouple was made of Pt-Rh10 wire and Pt wire as compensation wires, with a size of ≤ 1 mm and a thickness of about 6 μm. In the testing process, the static calibration of the thermocouple at 50~600 °C and 650~1500 °C was completed by a portable temperature verification furnace and a horizontal high temperature verification furnace, and the results showed that the Seebeck coefficient of the thermocouple was about 10.70 μV/°C, and its output voltage–temperature curve was similar to that of a standard S-type thermocouple, which achieved the effective temperature measurement up to 1500 °C. The dynamic response of Pt-Rh10/Pt thin-film thermocouple was then tested and studied using the pulsed laser method, and the results show that the time constant of the thermocouple prepared in this paper is about 535 μs, which has the characteristics of fast response and high precision high-temperature testing. Compared with the traditional thin-film thermocouple, the thermocouple has excellent electrical conductivity, more oxidation resistance, the surface layer is not easy to peel off and other advantages.

## 1. Introduction

With the development of information technology, micro-electromechanical system (MEMS) technology, new energy, new materials and other high-tech technologies, the fourth generation of initiating explosive device—MEMS-initiating explosive device, with the advantages of miniaturization, integration, multi-function, high precision and high reliability—has made great progress, which has a new requirement for dynamic measurement of the ignition temperature of initiating explosive devices [1,2]. The ignition temperature is an important index of the safety and performance of the explosive device, and the test and determination of the ignition point is an important basis for the reliability design, appraisal and evaluation of the explosive device [3,4,5].

For the measurement of the ignition temperature of the pyrotechnic agent, there are currently the equal heating rate method, and its improvement method, and the modeling simulation method. In the former, a quantitative sample is placed in a special test tube and heated at a specified heating rate [6]. At the same time, the tester visually observes the moment of ignition to determine the temperature of the heating medium when the sample burns or explodes, which represents the ignition point of the sample [7]. This method has a certain degree of danger and requires observation by human eyes to determine the ignition temperature of the sample, and there is a certain test deviation. At the same time, some researchers study the ignition temperature of the pyrotechnic agent through the method of micro-transducer modeling and simulation. Using the finite element analysis software ANSYS, through certain modeling assumptions, the finite element simulation model of the bridge wire or bridge mode electric pyrotechnic device is established, and the finite element analysis of the DC ignition process and the capacitor discharge ignition process of the electric pyrotechnic device is carried out [8]. The temperature change curve of the interface between the bridge wire or bridge film and the medicament (for example, chemically sensitive agents such as gunpowder) is obtained by simulation, and then the ignition temperature of the pyrotechnic device is obtained [9].

When the ammunition exploded, most of the heat released by the on-chip reactor was lost to the surrounding environment, and only a small part was detected. In 2013, Asaf Zuck et al. [10,11] found that the MEMS-based microcalorimetry can detect the melting and deflagration of particles with the same diameter as the explosive TNT. The ultra-thin thermocouple installed above the heater can detect heat loss and obtain temperature data that is highly consistent with the actual surface temperature. This research provides a basis for the use of MEMS technology to develop sensors for portable explosive detection. In the blasting field, the measurement of the generated high-temperature flowing fluid has always been an arduous task. In order to measure the static temperature, the measuring device should move with the fluid at the same speed without disturbing the flow rate, which is unrealistic. In 2015, Sonker et al. [12,13] of the Indian Institute of Technology, studied ultra-high temperature thin-film thermocouples based on this, which can be used in the maximum temperature range of 2900 K, and can be used to measure the heat flux of the missile and the gas flow through the nozzle. In 2016, Satish et al. [14] conducted basic experiments and proof-of-concept studies on a thin-film thermocouple deposited by the electron beam evaporation process. Its Seebeck coefficient was 42 μV/°C, and its time constant was 1.11784 ms. The proposed method can be applied in the temperature measurement of aviation gas turbine engines.

In summary, it is necessary to develop a tiny temperature measuring device to explore the ignition temperature test of pyrotechnic products. Through studying a large number of documents, it has good erosion resistance, low volume occupancy, high accuracy, short response time, etc. Thin-film thermocouples with advantages in temperature measurement can provide new ideas. Given the tiny junctions and size of the thin-film thermocouple, it has a small heat capacity with transient response [15]. Aiming at the current application environment of pyrotechnics’ ignition temperature testing methods, by studying materials with excellent performance and efficient thin-film thermocouple preparation methods, it can provide a new temperature measurement technology attempt for the ignition temperature testing of a new generation of pyrotechnics [16,17].

## 2. Process of Thermocouple Preparation

### 2.1. Structure

According to the general application scenarios of explosive devices, this paper selects a sheet structure to prepare a thin-film thermocouple. The structure has a smooth surface, which is conducive to better adhesion of the film layer, and can effectively fit with a variety of measured objects. The structure has high strength and good stability. The size of the micro energy commutator of the micro explosive device is generally in the millimeter level, so the size of the thin-film thermocouple developed should also be in the millimeter level. According to the working principle of a thermocouple, when two electrodes of a thermocouple contact each other, the contact potential difference will be generated in the contact area, and this contact end is the coupling junction of the sensor. At the same time, the size of the hot junction directly affects the heat capacity and response time of the thermocouple, and the response of the film thermocouple will be improved if the size of the coupling junction is controlled within 1 mm.

This article explores the various structures of thermocouples and designs the structure of the micro-sized thin-film thermocouple, as shown in Figure 1a.

It is determined that the designed thin-film thermocouple integrated with the micro-transducer is mainly composed of a substrate and a thermoelectrode. The thermoelectrode layer is composed of thermoelectrode 1 and thermoelectrode 2 arranged in mirror symmetry along the center line of the substrate. The first electrode and the second hot electrode constitute the thermal junction of the thermocouple through the overlapping area. Among them, the micro transducer adopts a serpentine structure, which is distributed in a serpentine shape at a distance of 200 μm from the thermal junction, and is mirror-symmetrical along the center line of the ceramic substrate. The pads are symmetrically distributed on both sides, and the three-dimensional structure is shown in Figure 1b. Among them, the thermoelectrode is 7000 μm long, 500 μm wide and 2 μm thick. The size of the even junction is 500 × 500 μm. The size of the pad is 3250 × 2000 μm, and the pad is used to connect the compensation wire.

### 2.2. Materials

High-purity alumina ceramics have many advantages, such as high melting point, thermal shock resistance, corrosion resistance, good wear resistance, high temperature stability and good bonding force with Pt/Rh alloy film thermoelectrode [18]. In addition, the insulation resistance of high-purity ceramics is not less than that of ceramic materials. In view of the many advantages of ceramic materials, this paper selects high-purity alumina ceramics (99 porcelain) as the substrate for preparing thin-film thermocouples, and its structural matrix parameter is 10 mm (length) × 8 mm (width) × 0.5 mm (thickness).

According to the basic working principle of thermocouples and the law of intermediate conductors, the Seebeck coefficient of thermoelectrode material is the only factor that affects the thermoelectric potential in the closed circuit of thermocouples, and the thermoelectrode material determines the performance of thin-film thermocouples. The thermocouple made of precious metals is often used in high temperatures and harsh environments. Pt-RH10/Pt thermocouple in the thermocouple series has the advantages of high accuracy, good stability, wide temperature measurement area, long service life and high temperature measurement upper limits [19] and is suitable for oxidation and inert atmosphere. The tungsten–rhenium thermocouple is a high melting point metal thermocouple developed gradually in order to meet the demand of temperature information measurement in high temperature environments. It is also an industrial thermocouple that can meet the demand of temperature measurement in high temperature environments above 1800 °C at present. It is widely used in aerospace, metallurgy, the nuclear industry and other high temperature industries. It has the advantages of a high melting point of alloy electrode material, high output thermoelectric potential, high sensitivity, low price, and so on.

By comparing the advantages and disadvantages of the two materials, the precious metal platinum–rhodium material is more expensive than tungsten–rhenium, and vulnerable to pollution, so the one-time investment is large; therefore, this paper has used, respectively, tungsten–rhenium materials (positive extreme W:Re = 95 wt%:5 wt%, the preparation of thin-film thermocouple with negative electrode W:Re = 74 wt%:26 wt%), platinum–rhodium alloy and pure platinum as electrode materials (positive extreme Pt-Rh10, negative extreme pure Pt) was explored [20].

As the resistance of the transducer element increases, the critical ignition voltage of the transducer element increases in the form of a power function. When the resistance of the transducer is the same, the common Ni-Cr, Pt, and Cr materials have lower ignition voltage. Taking into account many factors such as process difficulty, fabrication cost, and material properties, the authors finally chose Pt as the transducer material for the design and integrated production of the micro-initiator [21].

Synthesizing the process compatibility and performance requirements of thin-film thermocouples, the device structure material is determined: high-purity alumina ceramics (99 porcelain) as the matrix material and Pt metal as the energy-transducing element material. According to verification requirements and experimental conditions, W-5%Re/W-26%Re thin-film thermocouples and Pt-Rh10/Pt thin-film thermocouples were separately prepared on the substrate.

In order to understand the heat distribution of the designed structure, the Pt-Rh10/Pt thin-film thermocouple with a substrate size of 10 mm (length) × 8 mm (width) × 0.5 mm (thickness) and a junction size of (500 × 500 μm) is used as an example. Then, the corresponding element type and material properties are defined, as shown in Figure 2. Below, the author uses Comsol finite element analysis software to simulate the designed structure and pulse laser to excite the coupling junction to simulate the heat transfer of the structure. Given that the laser spot area is smaller than the area of the thermocouple junction, it radiates heat longitudinally from the central region and has a good response. Since the properties of the material in the simulation process are selected as the thermal physical parameters of the unit solid-like material, it will inevitably lead to some errors in the thermoelectric properties of the thin film thermocouple under the simulation conditions, but the variation pattern of its temperature profile is consistent with the conventional physical properties of thermocouples.

### 2.3. Preparation

The magnetron sputtering method is adopted, and two WRe alloy targets with a purity of 99.99% with different composition ratios are used as the target materials (W:Re-95 wt%:5 wt%; W:Re-74 wt%:26 wt%; Φ76.2 mm × 6.35 mm). High-purity Ar is used as the sputtering gas, and the flow rate in working condition is 400sccm; the target base distance is adjusted to 110 mm, and then W-5%Re and W-26%Re are prepared on an Al_2_O_3_ ceramic substrate with a size of 10 mm × 8 mm × 0.5 mm. The detailed process and effect of thin-film thermocouple prepared by magnetron sputtering are shown in Figure 3.

In order to not affect the electrical performance of the thin-film thermocouple, the tungsten–rhenium wire, which is the same as the electrode material, is used as the lead wire, and the tungsten paste is the welding material. When welding the lead, it was found that the bonding force between the tungsten paste and the pad was poor and easy to fall off. Then, the sample was placed in a tube furnace for high-temperature treatment; the holding temperature was 800 °C, the time was 1 h, and the heating rate was 10 °C/min. In order to prevent the tungsten–rhenium film from being rapidly oxidized, argon gas was filled in the furnace, and the flow rate of argon gas is 400 sccm. The surface morphology after heat treatment is shown in Figure 4 below.

It can be seen that during high-temperature sintering, the organic matter of the conductive tungsten paste will volatilize during the heating process, the volume shrinks and discolors, and the shape is dry. The prepared W-5%Re/W-26%Re thin-film thermocouple still underwent oxidation in the hot end region. The film changed from silver-white metallic luster to dark green, and the film adhesion was not high. After studying the relevant literature, it is found that the tungsten–rhenium film thermocouple deposited with the Al_2_O_3_ protective layer will also fail in a short time under high-temperature environments, and the film protection process is not yet mature.

Based on previous experiments, the authors explore a low-cost and efficient preparation process, using screen printing technology to prepare Pt-Rh10/Pt thin-film thermocouples suitable for oxidizing and inert atmospheres on ceramic substrates. The preparation process is shown below.

(1) The substrate is selected and wiped with alcohol for surface hydrophilization;

(2) The platinum–rhodium conductive paste is mixed with a fineness of <15 μm with an organic carrier to make a platinum–rhodium conductive micron paste, and the platinum–rhodium electrode (left electrode) mask is placed on the insulating substrate to form a printing plate;

(3) The platinum–rhodium conductor micron paste is stacked on the printing plate and moved and pressed with a scraper to make it leak through the image area of the mask and the pores of the screen to the surface of the substrate, thereby forming a platinum–rhodium electrode film;

(4) Multi-layer printing and reinforcement of platinum and rhodium electrodes are achieved by performing leveling, drying and heat treatment on the obtained platinum rhodium electrode film and then cooled down naturally;

(5) This is replaced with a platinum electrode (right electrode) mask and a platinum electrode slurry with a fineness of <10 μm is mixed with an organic carrier to make a platinum electrode micron slurry. Step (2) to step (3) are repeated to form a platinum electrode film; the film thickness is about 6 μm;

(6) Multi-layer printing and reinforcement of the platinum electrode, film leveling, drying and heat treatment of the obtained electrode, and then natural cooling, are performed;

(7) The sample was dried in an oven and then sintered at a high temperature in a muffle furnace at 1300 °C for four hours to complete the preparation of the sensor.

Figure 5 below shows the screen-printing stencil structure during the preparation process and the final finished film thermocouple.

In the screen-printing manufacturing process, sintering is a very important process. The film dried on the substrate must be sintered to have certain electrical properties. The sintering process is decisive for the properties and composition of the film, and its important condition is the sintering temperature. Only by sintering under the most suitable temperature and other conditions can the best properties of the materials used be obtained. After each electrode is printed, in order to make the film thickness more uniform and not easy to deform, the prepared film specimens are dried in a muffle furnace environment temperature of 200 °C for 20 min. During this period, most of the organic matter in the film will be slow. Slow volatilization enhances the adhesion of the film layer and the substrate. After the film is air-dried, the shape and performance of the dried film will not be affected when printing on the other side. At the same time, it also prevents the binder in the slurry from escaping during high-temperature sintering and causing the newly printed film to break.

The dried thin-film thermocouple is placed in a muffle furnace for high-temperature sintering. The sintering steps include three stages of uniform heating, heat preservation and natural cooling. For different slurries, the appropriate high-temperature sintering temperature will directly affect the adhesion between the film and the substrate and the stability of the film. Too low of a temperature will result in unstable film structure and decrease in electrical conductivity. Too high of a sintering temperature will cause the crystal particle size difference in the film composition to be too large, resulting in a small oxidation reaction. Similarly, the length of the holding time will also affect the sintering effect. Too long of a holding time will continuously increase the crystal grains in the film and increase the resistance. Too short of a holding time will cause insufficient reaction of the film and lose the sintering effect. In order to improve the efficiency of preparing thin-film thermocouples and prevent excessive volatilization of organic solvents in the film, a muffle furnace with a maximum firing temperature of 1800 °C is used to fire the thin-film thermocouples during the sintering process, as shown in Figure 6a below. The organic binder consists of resin, solvent and additives, of which the resin and solvent are ethyl cellulose and pine alcohol, respectively, and, finally, additives such as surfactants, leveling agents and defoamers are added according to actual needs. After many sintering experiments, the specific temperature rise rate when sintering the Pt-Rh10/Pt thin-film thermocouple is 7.2 °C, as shown in Figure 6b below; the holding temperature is 1300 °C, and the holding time is 1 h. After four hours of sintering, it is cooled naturally. Intervention with cool-down can easily form internal stress in the film, which affects the stability of the film thermocouple.

## 3. Test and Analysis

### 3.1. Thickness Test

In this paper, the authors used a P6 Stylus Profiler to perform thickness measurements on thin-film thermocouples. The use of a high-resolution step meter not only makes it easier and more intuitive to measure the film thickness, but also accurately characterizes the roughness of the film surface. The stylus is perpendicular to the substrate, and, by measuring the change in displacement of the stylus in the vertical direction as it crosses the thermocouple surface, the surface characteristics can be accurately characterized. The physical diagram of the P6 Stylus Profiler is shown in Figure 7 below, and the measurement results for the thin-film thermocouple are shown in Table 1 below.

The measurement results in the table above illustrate that the average values of errors for the thickness, electrode width and its length of the thin-film thermocouple are 3.5%, 1.4% and 10%, respectively. The average thickness error of the thin-film thermocouple is 3.5%, which is not much different from the actual design of the theoretical value, basically to meet the design requirements. The thickness of the thin-film thermocouple, although there are some places where there is a certain small deviation, belongs to the normal phenomenon. The small deviations occur because the printing process is subject to a small range of perturbations in the gas environment and the muffle furnace sintering temperature, all of which can cause irregular changes in the crystal particles in the film layer.

### 3.2. Static Test

In order to better evaluate the static performance index of the prepared Pt-Rh10/Pt thin-film thermocouple sensor, the Pt-Rh10/Pt thin-film thermocouple is statically calibrated according to the requirements of JJG542-1997 platinum rhodium–platinum thermocouple verification regulations. The electric potential value of the standard graduation table is compared with the thermocouple under test at different temperature points.

The basic composition of the thermocouple temperature measurement circuit is shown in Figure 8. It is composed of a thermocouple, a compensation wire, a connecting wire and a secondary instrument. Among them, T represents the temperature of the measuring terminal, T1 represents the temperature of the terminal, and T0 is the temperature of the reference terminal. When indexing and calibrating the thermocouple, the temperature of the reference junction should be maintained at 0 °C throughout the measurement process, that is, to ensure that the temperature of the cold junction is 0 °C, and the measured electromotive force (mV) is proportional to the temperature of the measurement junction.

Because the calibration temperature range is wide, the static calibration of the film thermocouple is carried out in different regions, and different calibration equipment is used in different temperature ranges. A portable temperature calibration furnace to calibrate the thermocouple in the range of 50~600 °C is used, and a temperature point is set every 50 °C from 50 °C. The switch of the zero-degree thermostat is turned on, and 3ml of alcohol are added to each zero-degree hole. When the temperature of the working area of the zero-degree thermostat drops to 0 °C, the constant-temperature indicator light starts to flash, and the zero-degree thermostat enters the constant temperature state. After about 10 min, the zero-degree thermostat can be used as the reference terminal temperature of the thermocouple. In order to prevent the two lead wires from being contacted by mistake and causing a short circuit, the lead wires are inserted into the high-temperature-resistant ceramic tube before calibration for physical and mechanical protection. The bonding method is used to clamp the extension cord between the high-temperature-resistant ceramic sheet of the same material as the substrate and the thin-film thermocouple pad to form a sandwich structure. The tungsten rhenium sheet and the ceramic sheet are bonded and fixed, and the tested thermocouple is placed in the hole inside; the output terminal is connected to a high-precision digital multimeter through a zero-degree thermostat for measurement. The temperature of the heat source is set at 50 °C. When the temperature reaches the set value, the error is constant within ±1 °C, and the temperature fluctuation is not greater than 0.05 °C, the thermoelectric potential value of the measured film thermocouple sensor is recorded in real time with a high-precision digital multimeter, and the experimental site is shown in Figure 9a below.

The high temperature calibration furnace system is used for calibration in the range of 600~1500 °C. The static test system of the horizontal high-temperature calibration furnace is shown in Figure 3b above. In order to prevent the thermocouple from being affected by slight oxidation in the high temperature section, the system is equipped with an argon gas bottle device; the flow rate of argon gas under working condition is 400 sccm. Similar to the calibration process of the low temperature section, the film thermocouple is fixed on the slide rail of the fast feed device during calibration, the output end is placed in the zero-degree thermostat through the compensation wire, one end of the compensation wire in the zero-degree thermostat is connected to the digital multimeter through the wire on the multimeter, and the output voltage value of the film thermocouple is measured. The temperature test range of the high-temperature verification furnace is set to 600~1500 °C. The high-temperature furnace is heated at a heating rate of 5 °C/min, and the voltage value of the digital multimeter is recorded every 50 °C. The high-temperature static test experiment site is shown in Figure 10 below.

In order to improve the accuracy of the data and reduce the influence of the environment on the experiment, the value of the high-precision digital multimeter is recorded in real time during the experiment, and the collected data is processed to eliminate gross errors and find the average value. The relationship curve between thermoelectric potential and temperature is drawn and compared with the standard thermocouple index table, as shown in Figure 11 below.

From Figure 11 above, it can be intuitively found that the output thermoelectric potential of the tested thermocouple in the low temperature section is close to the standard thermocouple value, and the error is small. There is a slight deformation around 1200 °C, and the overall potential-temperature curve drops around 1mV in the range of 1200 °C to 1500 °C, resulting in a deviation of about 50 °C between the measured temperature and the standard temperature. After taking out the calibrated thermocouple, it was found that the clip at the lead connection was oxidized and deformed in a high temperature environment for a long time. As shown in Figure 12a below, the performance of the thermocouple in the high temperature furnace cavity has a slight impact. After a lot of experimental analysis and improvement, the sensor structure connected by welding method was tested, which is shown in Figure 12b below.

By controlling the temperature of the calibration furnace, the thermocouple is calibrated every 50 °C rise in furnace temperature, and the final scatter plot of thermal potential versus temperature is obtained. In order to more intuitively reflect the static performance of the thermocouple, it is necessary to curve-fit the thermocouple calibration data and establish a mathematical model of temperature–thermoelectric potential relationship. This paper chooses the least squares method that can match the best function according to the square sum of the minimum error to curve fitting the static calibration data. The three experimental data and the average fitting curve are shown in Figure 13 below.

After fitting by the least squares method, the Seebeck coefficient, fitting equation, correlation coefficient R^2^ and nonlinear fitting error of the tested thin-film thermocouple are summarized and analyzed, as shown in Table 2 below.

In view of the slow cooling rate of the horizontal high temperature appraisal furnace and the long time of the single calibration experiment, this article compares and analyzes the data of the above three repetitive experiments, and it can be seen intuitively that the thermoelectric potential data obtained by the Pt-Rh10/Pt thin-film thermocouple connected by the welding method in multiple repeated experiments are highly consistent, with excellent repeatability and stability. In this paper, the maximum standard deviation of experimental data obtained by the range method is 0.0123. The maximum repeatability error of the Pt-Rh10/Pt thin-film thermocouple is 0.0816%, and its repeatability is about 99.9184%.

It can be seen from the fitting equation of the static calibration data that the developed thin-film thermocouple has similar high temperature performance to the traditional filament S-type standard thermocouple, and the linear correlation coefficient R2 is close to 1, although there are hysteresis, creep, friction, etc., and in an external environment in the range of 50~1500 °C, the average Seebeck coefficient of the film temperature sensor can reach 10.70 μV/°C, which is close to the sensitivity of a standard S-type thermocouple, and is not affected by the thickness of the film. The sensor has good repeatability. The temperature can be measured continuously for more than three hours under the environment of 1500 °C, which meets the needs of engineering testing.

### 3.3. Dynamics Test

In view of the high energy density of laser and easy precise control, it is often used as an ideal heat source. At the same time, thanks to the flat shape of the self-made thin-film thermocouple junction, the laser energy can be effectively absorbed by the sensor. In this paper, a laser method is used to conduct dynamic experiments on the sensor. 

The thermal and shock resistance of thin-film thermocouples cannot be fully compared with the thermocouple of the block structure; if the energy irradiated by the laser is too large, the film or substrate material may be damaged to varying degrees, so the appropriate laser power is key to the success of the experiment. The laser used in this paper is the RFL-A500D semiconductor laser produced by Wuhan Rike Company, and the key technical parameters of the laser are shown in Table 3 below.

The IGA 740-LO infrared thermometer with a response time of 6μs is used to measure the surface pulse temperature of the thin-film thermocouple. The infrared thermometer has an adjustable emissivity, and the measured temperature signal can be used as the temperature excitation signal of the sensor.

First, the film sensor is fixed on the lifting platform with the aid of the infrared rays provided by the laser. First, the height of the lifting platform is adjusted so that the focus of the infrared light falls roughly within the temperature sensing area of the film thermocouple, and then the position of the optical lens is adjusted to adjust the focal length to ensure that the laser focus is the smallest and aligned with the film thermocouple, the center of the hot junction. The Pt-Rh10/Pt thin-film thermocouple is connected to the signal amplifier circuit through the extension cable; the amplified signal and the output signal of the infrared thermometer are input to the 8-channel high-speed synchronous data acquisition instrument. The laser switch is controlled, the signal of the film thermocouple and the signal of the infrared thermometer are synchronously collected. The laser power is set by the laser host computer software. In the Pt-Rh10/Pt dynamic test experiment, the laser power is set to 60%, 70%, 80% and 90% of the highest power of the laser, and the laser output time is 1 ms. The experimental site of the dynamic test of the thin-film thermocouple is shown in Figure 14 below.

For temperature sensors such as thermocouples, the time constant is the most important dynamic parameter, which reflects how fast or slow the thermocouple responds in dynamic measurements. The time constant is defined as the time it takes for the response of a temperature sensor under a temperature step excitation to reach 63.2% of its stable value from the starting moment. The time constant of the Pt-Rh10/Pt thin-film thermocouple was measured under different laser powers, as shown in Table 4 below, and the table also gives the excitation temperature and the highest temperature of the thermocouple under different laser powers.

In view of the thickness of the prepared thin-film thermocouple being 6 μm, the junction area is ≤1 mm and the surface is flat; the absorption rate of the laser is lower, and the time constant can reach the order of microseconds. It can be seen from the table that the average time constant of the Pt-Rh10/Pt thin-film thermocouple is about 535 μs.

## 4. Conclusions

Aiming at dynamic testing of the ignition temperature of micro-initiating explosives, a temperature sensor with high temperature stability and fast dynamic response was prepared. The preparation method of thin-film thermocouple was explored, and W-5%Re/W-26%Re thin-film thermocouple was prepared on Al_2_O_3_ ceramic substrate by magnetron sputtering method. The focus is on the preparation and performance research of Pt-Rh10/Pt thin-film thermocouples. The Pt-Rh10/Pt thin-film thermocouples with coupling size ≤1 mm are fabricated on Al_2_O_3_ ceramic substrates by screen printing. The results of fitting the thermocouple static calibration data reveal that the Seebeck constant of the thermocouple is 10.70 μV/°C, and it has good repeatability and stability. From the data in Table 4 above, it can be analyzed that the time constant of the thermocouple measured when the laser power is between 60% and 90% is about 535 μs, on average. It overcomes the connection problem between the thin-film thermocouple and the extension wire under the high temperature environment of 1500 °C in the static calibration test and has better conductivity, oxidation resistance, and it is not easy for the film to fall off. It is believed that the film prepared by this process has the potential for long-term application in a high temperature environment of 1500 °C.

## Figures and Tables

**Figure 1 micromachines-14-00004-f001:**
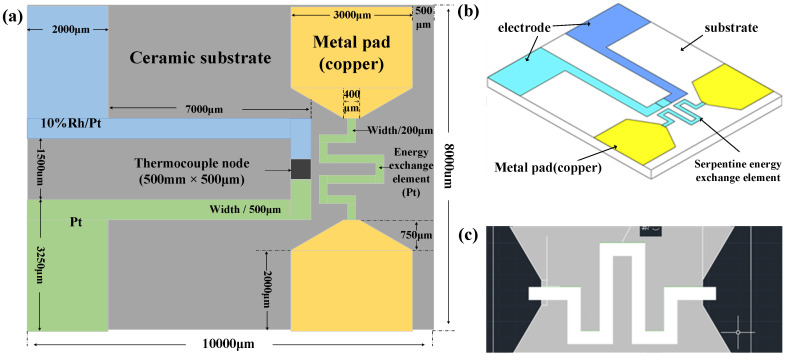
Structure diagram of Pt-Rh10/Pt thin-film thermocouple. Thin-film Thermocouple Structure Diagram: (**a**) Thin-film thermocouple plan structure diagram; (**b**) Thin-film thermocouple three-dimensional structure diagram; (**c**) Serpentine micro-energy exchange element structure diagram.

**Figure 2 micromachines-14-00004-f002:**
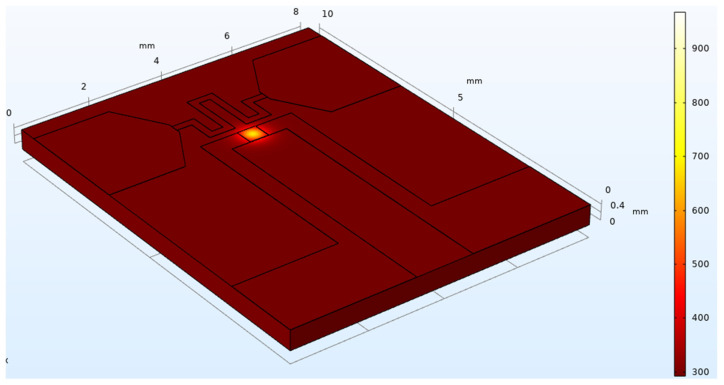
The temperature distribution of thermocouple is simulated by Comsol software.

**Figure 3 micromachines-14-00004-f003:**
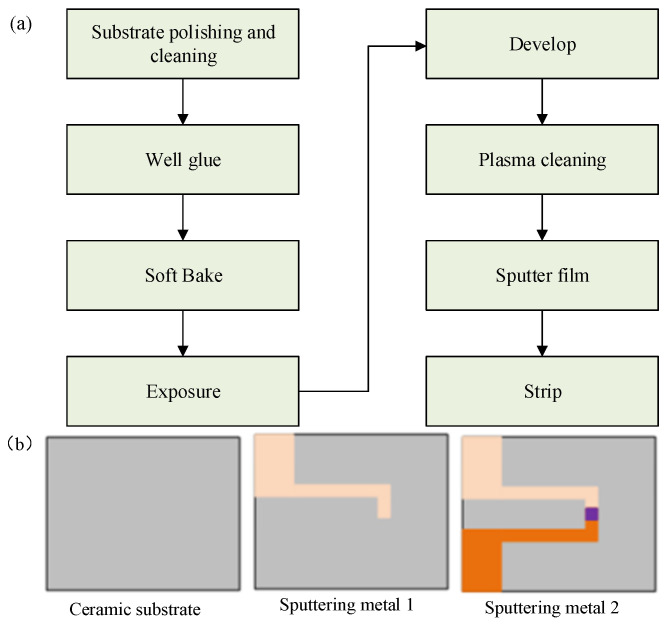
The detailed process and effect drawing of thin-film thermocouple prepared by magnetron sputtering method: (**a**) Diagram of sputtering process; (**b**) Ceramic base and two different sputtered metals.

**Figure 4 micromachines-14-00004-f004:**
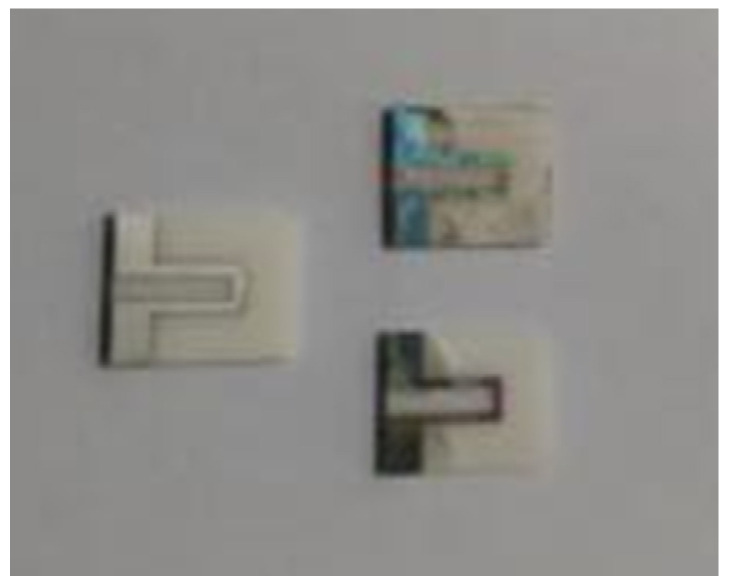
Tungsten–rhenium thermocouple after high temperature oxidation.

**Figure 5 micromachines-14-00004-f005:**
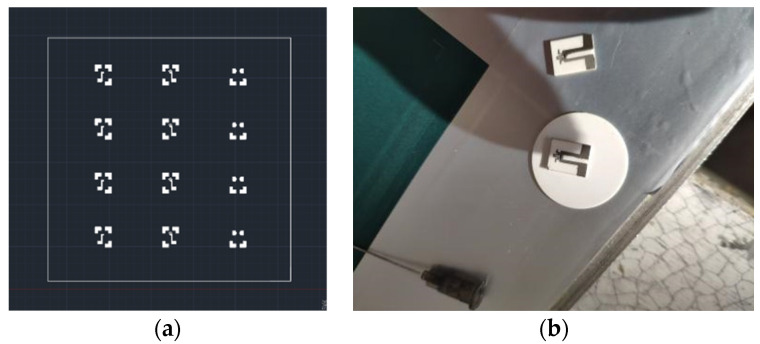
Screen-printing screen structure during preparation and final thin-film thermocouple products: (**a**) Screen-printing screen structure; (**b**) The thermocouple sample has been made.

**Figure 6 micromachines-14-00004-f006:**
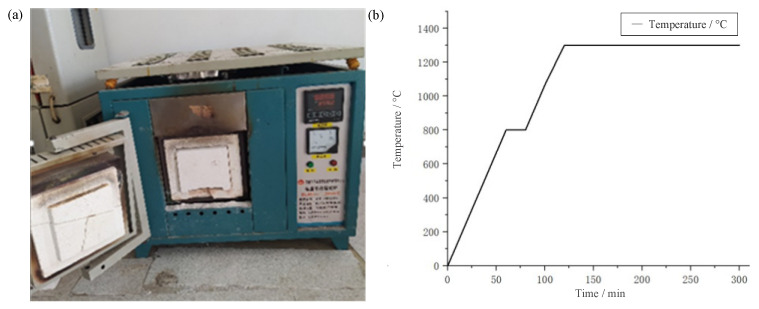
High temperature sintering of thermocouples by means of a muffle furnace: (**a**) Muffle furnace used in the experiment; (**b**) The temperature variation of Muffle furnace in the preparation.

**Figure 7 micromachines-14-00004-f007:**
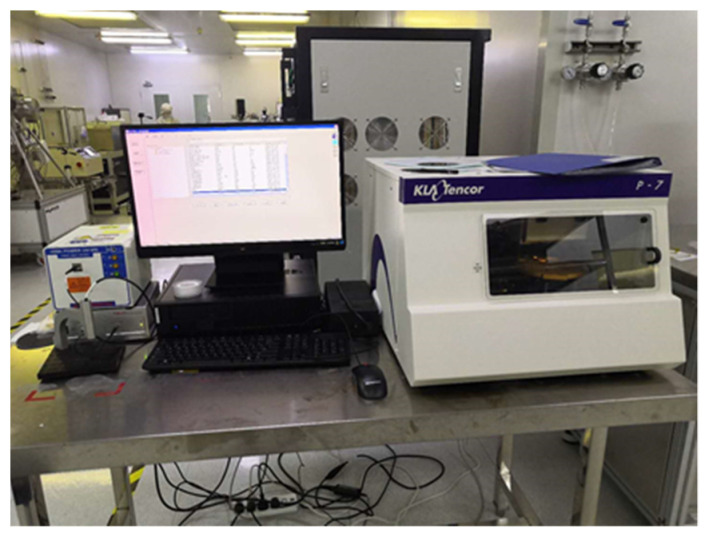
Physical view of P6 Stylus Profiler.

**Figure 8 micromachines-14-00004-f008:**
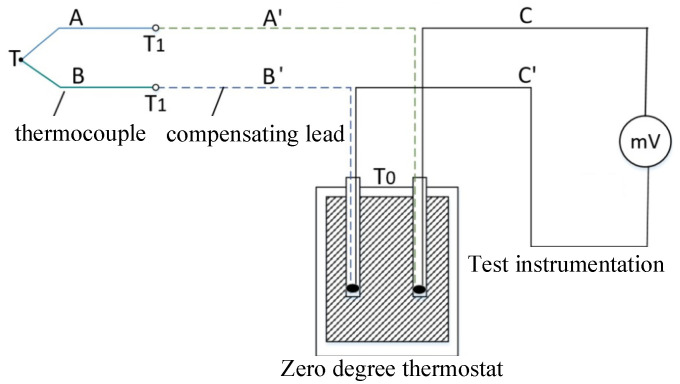
The basic composition of the thermocouple temperature measurement circuit.

**Figure 9 micromachines-14-00004-f009:**
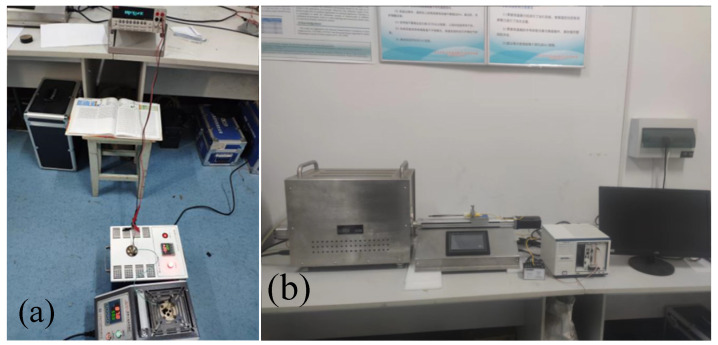
Static test system: (**a**) Low temperature section static test site; (**b**) Horizontal high temperature verification furnace static test system.

**Figure 10 micromachines-14-00004-f010:**
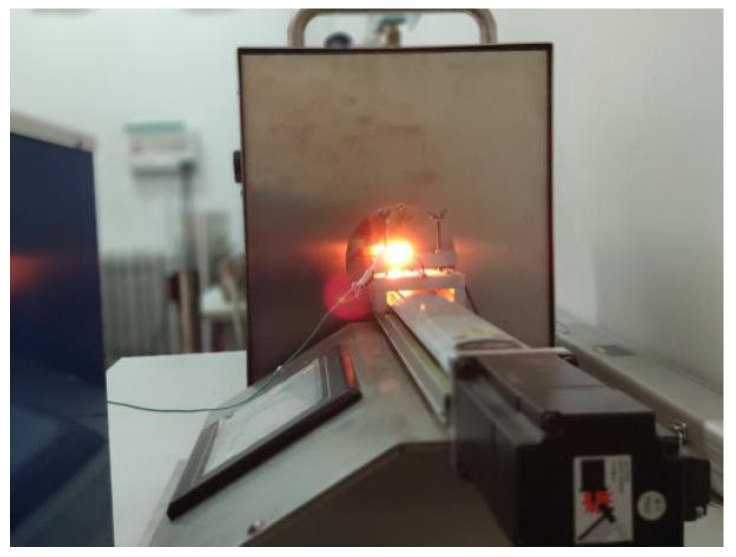
Thin-film thermocouple high-temperature section static test experiment site.

**Figure 11 micromachines-14-00004-f011:**
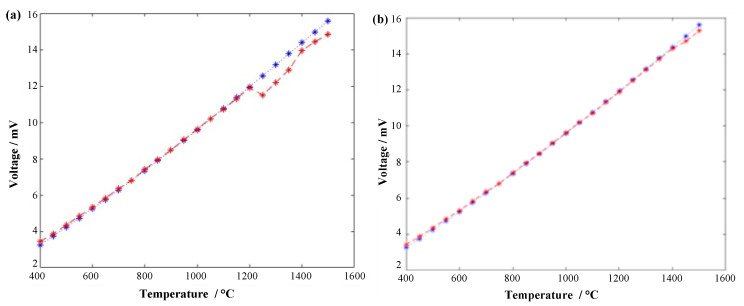
Comparison of the thermopotential versus temperature curves of thin-film thermocouples and standard thermocouples: (**a**) Fit of test data of thermocouples connected by keying; (**b**) Fit of thermocouple test data connected by soldering method.

**Figure 12 micromachines-14-00004-f012:**
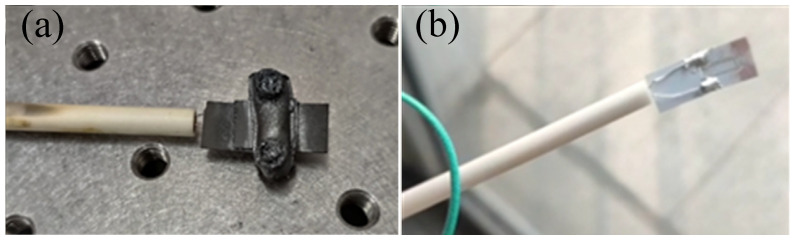
Oxidation deformation state of thermocouple under prolonged high temperature environment: (**a**) Ceramic bonded sheet after high temperature calibration; (**b**) Sensor structure for lead connection by welding.

**Figure 13 micromachines-14-00004-f013:**
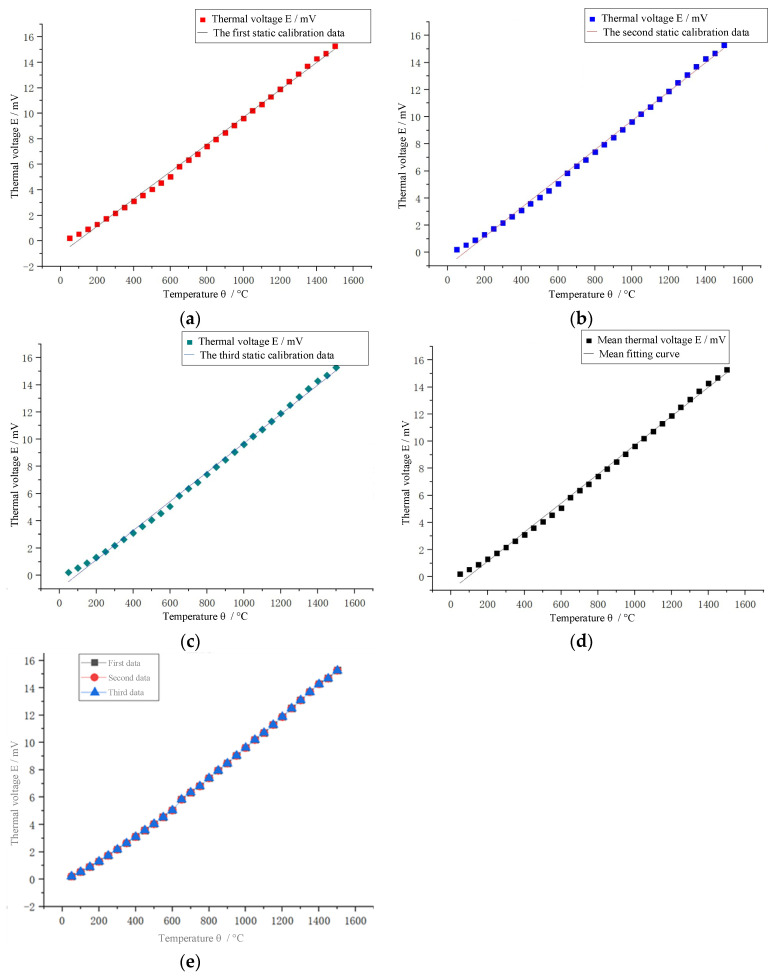
Data fitting of thin-film thermocouple after three calibration experiments: (**a**) First static calibration data fitting curve; (**b**) Second static calibration data fitting curve; (**c**) Third static calibration data fitting curve; (**d**) Mean fit curve for static calibration data; (**e**) Repeatability of data from three calibration experiments.

**Figure 14 micromachines-14-00004-f014:**
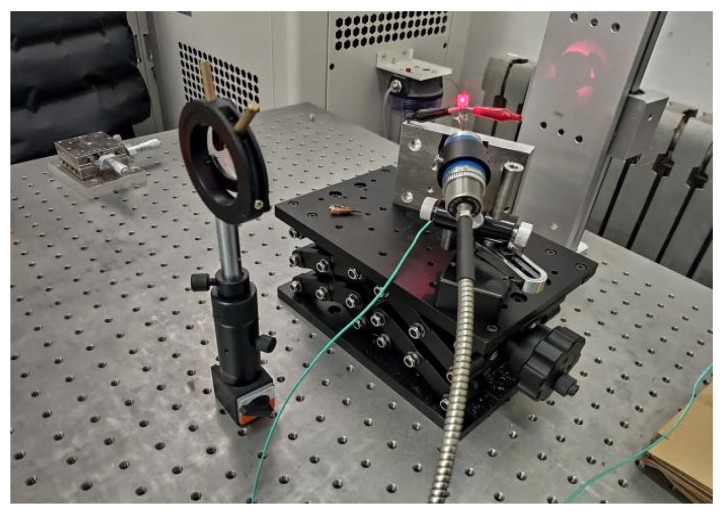
Thin-film thermocouple dynamic test experiment site.

**Table 1 micromachines-14-00004-t001:** The measurement results for the thin-film thermocouple.

Measurement Items	Measured Value	Theoretical Value	Error Value
Thermocouple thickness	6.21 μm	6.0 μm	3.5%
Electrode length	7.01 mm	7.0 mm	1.4%
Electrode width	550 μm	500 μm	10%
Hot node area	0.52 × 0.51 mm^2^	0.5 × 0.5 mm^2^	6.1%

**Table 2 micromachines-14-00004-t002:** Thermocouple static calibration data fitting results.

Times	Fitted Equation	Seebeck Coefficient (µV/°C)	Standard Error of Slope	Coefficient of Association (R^2^)
1	*E* = 0.01071*θ* − 0.999	10.71	0.01042%	0.99726
2	*E* = 0.01070*θ* − 0.997	10.70	0.01041%	0.99726
3	*E* = 0.01070*θ* − 0.996	10.70	0.01043%	0.99725
average	*E* = 0.01070*θ* − 0.997	10.70	0.01042%	0.99726

**Table 3 micromachines-14-00004-t003:** List of key technical parameters of the laser.

**Rated Power (W)**	**Power Adjustment Range (%)**	**Central Wavelength (nm)**	**Output Power Instability**
500	10~100	915 ± 10	<3%
**Output Head Type**	**Modulation Frequency (Hz)**	**Fiber Core Diameter (μm)**	**Beam Divergence Angle (Rad)**
QBH	50~20 k	300	<0.22

**Table 4 micromachines-14-00004-t004:** Time constants of thin-film thermocouple at different laser power.

Laser Power Percentage	Maximum Temperature (°C)	Maximum Excitation Temperature (°C)	Intrinsic Time(μs)
60%	466.3	661.3	600
70%	531.1	741.7	610
80%	642.5	881.6	490
90%	838.2	1069.0	440

## Data Availability

The data that support the findings of this study are available from the corresponding author upon reasonable request.

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
