# Peer review of "Fabrication and Calibration of Pt-Rh10/Pt Thin-Film Thermocouple"

_micromachines, 2022, doi:10.3390/mi14010004_

Round 1
Reviewer 1 Report
The authors carried out the preparation of the Pt-Rh10/Pt thin-film thermocouple by using a screen printing process for high temperature measurement, and conducted the static and dynamic test of the thermocouple. I recommend to publish the paper after the following issues are revised:
1. The authors used platinum-rhodium conductive paste with a fineness of <15μm and platinum electrode slurry with a fineness of <10um to form Pt-Rh10/Pt films by screen printing, respectively. How to assure the thickness of the films is exactly about 6 μm? [Page 6]
2. Please clarify that the air-dried process can prevent the organic binder in the slurry from escaping during high-temperature sintering of 1300℃. What composition is the organic binder?[Page 7]
3. In Fig. 12, for comparation, please provide the test data of thermocouple connected by welding method.
4. Line 348, what is the meaning of “Since calibration is performed every 50°C isolation”?
5. Line 358, correlation coefficient R2 should be R2? Table 1 should be also corrected.
6. Fig. 14, it is better that the four calibration data are placed into one picture to show the high repeatability.
7. Fig. 16 shows a response curve to obtain the time constant of the sensor. Please explain how to get the value of the time constant from the curve.
Author Response
Manuscript ID: micromachines-2066658 - Major Revisions
Original Article Title: “Fabrication and calibration of Pt-Rh10/Pt thin-film thermocouple”
Dear Editor,
We are honored to submit our manuscript in your journal and thank you for allowing us to revise the manuscript and giving us the opportunity to respond to the reviewers' comments. Next, we will respond to each of the two reviewers' and the editor's guidelines through revisions.
- Guidance:The authors used platinum-rhodium conductive paste with a fineness of <15μm and platinum electrode slurry with a fineness of <10um to form Pt-Rh10/Pt films by screen printing, respectively. How to assure the thickness of the films is exactly about 6μm? [Page 6]
Modification:The author has added section 3.1 to the paper to better describe how to test the thickness of thermocouples, etc. This section describes how to test the thickness of thermocouples with the P6 Stylus Profiler. [See Page 7, Line 303 for details.]
- Guidance:Please clarify that the air-dried process can prevent the organic binder in the slurry from escaping during high-temperature sintering of 1300℃. What composition is the organic binder?[Page 7]
Modification:The organic binder consists of resin, solvent and additives, of which the resin and sol-vent are ethyl cellulose and pine alcohol respectively, and finally additives such as surfactants, leveling agents and defoamers are added according to actual needs. [See Page 7, Line 290 for details.]
- Guidance:In Fig. 12, for comparation, please provide the test data of thermocouple connected by welding method.
Modification:The author has added Figure (b) after Figure 12 in the article (the icon has been changed to Figure 11 after all revisions of the article), which provides the test data of thermocouple connected by welding method. [See Page 10, Line 404 for details.]
- Guidance:Line 348, what is the meaning of “Since calibration is performed every 50°C isolation”?
Modification:It means “By controlling the temperature of the calibration furnace, the thermocouple is calibrated every 50°C rise in furnace temperature, and the final scatter plot of thermal potential versus temperature is obtained”. In addition, the content of the place has been retold in the article. [See Page 11, Line 432 for details.]
- Guidance:Line 358, correlation coefficient R2 should be R2? Table 1 should be also corrected.
Modification:In the article, these two errors have been corrected. [See Page 12, Line 453&456 for details.]
- Guidance: 14, it is better that the four calibration data are placed into one picture to show the high repeatability.
Modification:In the article, Figure (e) has been added to Figure 14 (the icon has been changed to Figure 13 after all revisions of the article), which shows the reproducibility of the data for the three calibration experiments. [See Page 12, Line 448 for details.]
- Guidance: 16 shows a response curve to obtain the Ome constant of tha sensor Please expiain how to get the value oj the time constant from the curve.
Modification:The time constant is defined as the time it takes for the response of a temperature sensor under a temperature step excitation to reach 63.2% of its stable value from the starting moment. However, the author feels that Figure 16 not only fails to express what the author wants to express, but also may cause distress to the reader, so the author feels that this picture should not exist. So in the article the author removed this picture and through Table 2 was also able to express the finding of the time constant for the thin film thermocouple. [See Page 14, Line 530 for details.]
We would be very grateful if you could assign our article to the same reviewers.
Thanks.
Best regards,
Fengxiang Wang,Zhijie Zhang et al.
E-mail address: 3371750184@qq.com
zhangzhijie@nuc.edu.cn

Reviewer 2 Report
Dear Editor,
I reviewed the article “Fabrication and calibration of Pt-Rh10/Pt thin-film thermocouple”
This article shows interesting performances for temperature measurement by miniaturized thermocouples.
Important remarks:
Fig 16 : The format is not acceptable, also the exposure for the time seems not be in adequacy with the Table 2 for the time. It must be absolutely corrected or explained.
The article should show some additional information as microstructure of the contact at the junction. As example a cross section by SEM could be very profitable. (Evolution of the junction with the temperature=> Life time of the sensor
I have some intermediate remarks
Typography of > symbol is not ok
A lot of figure could be combined (Fig1 and 2) as example, Fig 7 and 8.
L56 : What is medicament ? I have not understood what is it here!
L81 : Its smaller heat capacity (in fact it is both the matter quantity and heat capacity) which limit the heat pumping (which need to be as low as possible)
L 151 : Define the ignition voltage ? What is it
L173 : the pressure is not well expressed (must be corrected)
There are a too much decimal for value (A better exposure of the results with metrological consideration must be used).
Ref 6 : Change al/mno2 by Al/MnO2
May be replace static by steady test.
May be reduce some experimental detail (or to add a supplementary information) for (L206-L225)
What is the value of the absorbance at the laser wave length (optical data should be added if possible)?
Minor remarks
L195: . The hot… it lacks a point.
L318: On must be changed in on.
L359 : R2 must be changed into R²
L91 : Conducive must be replaced by conducting
I suggest Major revision, but the article is well adapted to the Journal and give interesting results.
Sincerely
Author Response
Manuscript ID: micromachines-2066658 - Major Revisions
Original Article Title: “Fabrication and calibration of Pt-Rh10/Pt thin-film thermocouple”
Dear Editor,
We are honored to submit our manuscript in your journal and thank you for allowing us to revise the manuscript and giving us the opportunity to respond to the reviewers' comments. Next, we will respond to each of the two reviewers' and the editor's guidelines through revisions.
- Guidance:Fig 16: The format is not acceptable, also the exposure for the time seems not be in adequacy with the Table 2 for the time. It must be absolutely corrected or explained.
Modification:This review comment is similar to the previous one and will not be repeated for the sake of space.
- Guidance:The article should show some additional information as microstructure of the contact at the junction. As example a cross section by SEM could be very profitable. (Evolution of the junction with the temperature=> Life time of the sensor)
Modification:The author would love to observe the microstructure of the thermocouple by means of SEM scanning electron microscopy, etc., but unfortunately the laboratory conditions are limited and there is a lack of relevant equipment. However, the author has submitted a request to his supervisor to procure more specialized equipment for subsequent in-depth studies.
- Guidance:Typography of > symbol is not ok
Modification:This inappropriate notation has been modified in the article. [See Page 1, Line 18 for details.]
- Guidance:A lot of figure could be combined (Fig1 and 2) as example, Fig 7 and 8.
Modification:Fig1 and Fig2, and Fig 7 and Fig8 have been merged in the article. In addition, inappropriate figures and tables have been removed or added accordingly according to the expert's guidance. Finally, the author has renumbered the figures and tables. [See Page 3, Line 110 and Page 7, Line 300 for details.]
- Guidance:What is medicament? I have not understood what is it here!
Modification:The medicament here refers specifically to chemically sensitive agents in the field of pyrotechnics, the most familiar type of agent is gunpowder. [See Page 2, Line 57 for details.]
- Guidance:Its smaller heat capacity (in fact it is both the matter quantity and heat capacity) which limit the heat pumping (which need to be as low as possible)
Modification:The expert's guidance is correct. The advantage of thin film thermocouple is that its thermal contact can be made very small, with a small heat capacity and fast response time, which is very suitable for surface temperature on a tiny area and dynamic temperature measurement of rapid changes. [See Page 2, Line 82 for details.]
- Guidance:Define the ignition voltage? What is it?
Modification:Ignition voltage is a common concept in pyrotechnics. First of all, we need to understand the concept of ignition circuit. The electrolytic capacitor with energy storage and the small ohmic resistor form a discharge circuit, which will be broken down and discharged when the ignition voltage value is reached at both ends of the capacitor. The instantaneous high current generated can produce high heat in the resistor, which is enough to cause the sensitive chemicals coated on the resistor to fire, and such a circuit is a typical pyrotechnic ignition circuit. The value of the voltage that causes the capacitor to be broken down is the ignition voltage.
- Guidance:the pressure is not well expressed (must be corrected)
Modification:The author's statement here in the original text does contain an obvious error and a correction has been made: “Use high-purity Ar as the sputtering gas and the flow rate in working condition is 400sccm, adjust the target base distance to 110mm, and then prepare W-5%Re and W-26%Re on an Al2O3 ceramic substrate with a size of 10mm×8mm×0.5mm”. [See Page 4, Line 187 for details.]
- Guidance:There are a too much decimal for value (A better exposure of the results with metrological consideration must be used).
Modification:In the article, the author has streamlined the number of decimal places without affecting the expressed results. [See Page 12, Line 456 for details.]
- Guidance:Ref 6: Change al/mno2 by Al/MnO2
Modification:The error that occurred there has been corrected in the article. [See Page 15, Line 591 for details.]
- Guidance:May be replace static by steady test.
Modification:The authors believe that both "static test" and "steady test" can be used, but in order to echo and contrast with "dynamic test" later, it is more appropriate to use "static test" which is most often used in the domain of testing.
- Guidance:What is the value of the absorbance at the laser wave length (optical data should be added if possible)?
Modification:In response to the expert's important guidance, the author has added a section to the article (see page 13 for details) that describes the RFL-A500D semiconductor used in the test and lists some key technical parameters. [See Page 13, Line 494 for details.]
- Guidance:L195: The hot… it lacks a point.
Modification:The error that occurred there has been corrected in the article. [See Page 5, Line 219 for details.]
- Guidance:L318: On must be changed in on.
Modification:The error that occurred there has been corrected in the article. [See Page 10, Line 392 for details.]
- Guidance:L359: R2 must be changed into R²
Modification:The error that occurred there has been corrected in the article. [See Page 12, Line 453&456 for details.]
- Guidance:L91: Conducive must be replaced by conducting
Modification:The error that occurred there has been corrected in the article. [See Page 2, Line 92 for details.]
We believe that we have adequately addressed all the points the editor and reviewers have raised and the paper is significantly improved.
We would be very grateful if you could assign our article to the same reviewers.
Thanks.
Best regards,
Fengxiang Wang,Zhijie Zhang et al.
E-mail address: 3371750184@qq.com
zhangzhijie@nuc.edu.cn

Round 2
Reviewer 2 Report
Dear Editor,
I have a last remark about the COMSOL simulation in the Fig 2. Why temperature is very different at the Thermocouple junction and the serpentine? The simulation is devoted to hot laser spot which generate a radial temperature gradient. Due to this difference of temperature between junction and serpentine, it may generate a static error for the true temperature determination. It would be very profitable to add one sentence to explain this point to the reader.
L490 : The time constant is about It is 535μs and the Seebeck coefficient is about 10.70μV/°C.
Sentence must be corrected.
Excepted these remarks, I agree with the article which can be published in your Journal
Sincerely
Author Response
Manuscript ID: micromachines-2066658 - Major Revisions
Original Article Title: “Fabrication and calibration of Pt-Rh10/Pt thin-film thermocouple”
Dear Editor,
We are honored to submit our manuscript in your journal and thank you for allowing us to revise the manuscript and giving us the opportunity to respond to the reviewers' comments. Next, we will respond to each of the two reviewers' and the editor's guidelines through revisions.
- Guidance:The editor sent an email and mentioned in it that the email addresses of the respective authors were not appropriate.
Modification:Changes have been made to each of the four authors' inappropriate email addresses.
- Guidance:I have a last remark about the COMSOL simulation in the Fig 2. Why temperature is very different at the Thermocouple junction and the serpentine? The simulation is devoted to hot laser spot which generate a radial temperature gradient. Due to this difference of temperature between junction and serpentine, it may generate a static error for the true temperature determination. It would be very profitable to add one sentence to explain this point to the reader.
Modification:Given that the laser spot area is smaller than the area of the thermocouple junction, it radiates heat longitudinally from the central region and has a good response. Since the properties of the material in the simulation process are selected as the thermal physical parameters of the unit solid-like material, it will inevitably lead to some errors in the thermoelectric properties of the thin film thermocouple under the simulation conditions, but the variation pattern of its temperature profile is consistent with the conventional physical properties of thermocouples. [See Page 4, Line 187 for details.]
- Guidance:L490 : The time constant is about It is 535μs and the Seebeck coefficient is about 10.70μV/°C.
Modification:The meaning expressed in this sentence is indeed very vague and incomplete, and the author has corrected it. [See Page 14, Line 555 for details.]
We would be very grateful if you could assign our article to the same reviewers.
Thanks.
Best regards,
Fengxiang Wang, Zhijie Zhang et al.
E-mail address: wangfengxiang123@126.com
zhangzhijie@nuc.edu.cn
